# Ethylcellulose in Organic Solution or Aqueous Dispersion Form in Designing Taste-Masked Microparticles by the Spray Drying Technique with a Model Bitter Drug: Rupatadine Fumarate

**DOI:** 10.3390/polym11030522

**Published:** 2019-03-20

**Authors:** Katarzyna Wasilewska, Marta Szekalska, Patrycja Ciosek-Skibinska, Joanna Lenik, Anna Basa, Julia Jacyna, Michal Markuszewski, Katarzyna Winnicka

**Affiliations:** 1Department of Pharmaceutical Technology, Medical University of Białystok, Mickiewicza 2c, 15-222 Białystok, Poland; katarzyna.wasilewska@umb.edu.pl (K.W.); marta.szekalska@umb.edu.pl (M.S.); 2Departament of Microbioanalytics, Warsaw University of Technology, Koszykowa 75, 00-662 Warsaw, Poland; pciosek@ch.pw.edu.pl; 3Department of Analytical Chemistry and Instrumental Analysis, Faculty of Chemistry, Maria Curie-Skłodowska University, M. Curie-Skłodowska Sq. 3, 20-031 Lublin, Poland; j.lenik@poczta.umcs.lublin.pl; 4Institute of Chemistry, University of Białystok, Ciołkowskiego 1K, 15-245 Białystok, Poland; abasa@uwb.edu.pl; 5Department of Biopharmaceutics and Pharmacodynamics, Medical University of Gdańsk, Hallera 107, 80-416 Gdańsk, Poland; julia.jacyna@gumed.edu.pl (J.J.); michal.markuszewski@gumed.edu.pl (M.M.)

**Keywords:** ethylcellulose, taste masking polymer, cellulose derivative, spray drying, microparticles, rupatadine fumarate

## Abstract

The taste of drugs is an important factor affecting pharmacotherapy effectiveness, and obtaining formulations with acceptable organoleptic properties is still an ongoing issue in pharmaceutical technology. One of the innovative methods of taste masking is preparation of microparticles by the spray drying technique, utilizing polymers with different physicochemical properties. Rupatadine fumarate (RUP) is one of the newest antihistamines, with an innovative and multidirectional mechanism of action, and an extremely bitter taste. The aim of this work was to investigate the feasibility of utilizing organic or aqueous forms of ethylcellulose (EC) for the preparation of microparticles with RUP by the spray drying technique. Spray dried samples at different drug:polymer ratios were prepared using organic solution (Ethocel^®^) or aqueous dispersions of EC (Surelease^®^, Aquacoat^®^ ECD). Evaluation of the taste masking efficacy was performed in vivo in human taste panel, in vitro based on dissolution test, and by self-constructed electronic tongue. It was shown that microparticles obtained from aqueous dispersions of EC have superior pharmaceutical properties in terms of both morphology and taste masking efficacy in comparison to those obtained from organic solution.

## 1. Introduction

Taste is the sensation that results when taste buds in the tongue and throat convey information about the chemical composition of a soluble stimulus [1]. Due to the fact that medicines taken by a patient have direct contact with the buds, taste and palatability are crucial factors influencing compliance and adherence [2]. A variety of taste masking methods, such as addition of flavor enhancers (sucrose, mannitol, sorbitol, aromas), drug solubility modification, complexation, e.g., with cyclodextrins, or microparticle preparation can be distinguished. Microparticles are considered to be the basis for preparation of various dosage forms, including modified-release tablets or capsules, as well as orodispersible tablets and films. Microparticles are used to improve drug absorption and durability, modify drug release, and mask unpleasant tastes [3,4]. Preparing microparticles using taste masking polymers is considered to be the most successful method for reducing the bitterness of a drug. In microparticle preparation, both hydrophilic and hydrophobic polymers are utilized. One of the methods of obtaining microparticles is spray drying, which has been successfully used as an effective and efficient technological platform [5]. Ethylcellulose (EC)—derived from cellulose in the ethyl ether form—is a polymer applied as a taste and smell masking agent [6]. The European Pharmacopoeia (Ph. Eur.) and United States Pharmacopoeia (USP) monographs describe EC as partly O-ethylated cellulose, containing not less than 44% and not more than 51% of ethyl groups (-OC_2_H_5_). It exhibits a high degree of stability [7,8]. EC is a non-toxic (generally recognized as safe—GRAS), non-irritant, colorless, odorless, and biocompatible polymer. It is compatible with a wide array of excipients and most of the plasticizers used in pharmaceutical formulations [9]. The Food and Drug Administration (FDA) included this polymer in its inactive ingredient database [10]. EC is widely used in the pharmaceutical technology, mostly in oral formulations. It is primarily employed as a coating agent modifying drug release, improving drug stability, or providing moisture protection. EC is also utilized in modified-release tablets as an excipient creating a matrix structure. Commercially, it is available in a wide range of viscosity grades as well as in the form of aqueous dispersions, giving a good range of possibilities for the formulators [11,12,13,14,15,16]. Examples of solid organic forms are Ethocel^®^ and Aqualon^™,^ and of aqueous dispersions Aquacoat^®^ ECD, Surelease^®^, Aquarius^™^ Control ECD, or AshaKote^®^ [17,18,19,20,21]. Ethocel^®^ is produced in two ethoxyl types and in a number of different viscosities [17]. Aquacoat^®^ ECD is a 27% dispersion of EC. It is prepared by an emulsification solvent evaporation method, where the system is stabilized with sodium lauryl sulfate and cetyl alcohol [19]. Surelease^®^ has a 25% solid content of EC, and it is plasticized with oleic acid, dibutyl sebacate, or fractionated coconut oil (depending on type) [20]. It is produced by blending EC with plasticizer, followed by melting and extrusion. Next, the mixture is directly emulsified in ammoniated water under pressure. Various grades of Surelease^®^, with differences in medium-chain triglycerides, dibutyl sebacate, and colloidal silica content, are available [20]. EC creates an effective barrier against the movement of drug molecules to the surface and water molecules to the core, thus providing a taste masking effect [6,22]. 

As a model bitter-tasting drug, RUP, one of the newest antihistamines, with anti-allergic and anti-inflammatory action, was used. RUP is a potent, selective antagonist of histamine receptors H_1_ and receptors for platelet activating factor (PAF), which specifically distinguishes RUP from other antihistamines and explains the innovative mechanism of its anti-inflammatory action. Comparative studies have shown that RUP has the highest affinity for the H_1_ receptors among antihistamines—significantly greater than fexofenadine or levocetirizine. RUP also binds to receptors for PAF (a potential marker of anaphylaxis), causing their blockade, and decreases platelet aggregation [23,24]. Because RUP is characterized by an extremely bitter taste, a technological challenge is leveling unpleasant feelings, simultaneously giving an opportunity of designing novel, patient friendly orodispersible formulations, which are currently lacking in the pharmaceutical market. 

Despite widespread utilization of EC in the pharmaceutical technology [6,11,12,13,14,15,16,25], there are only a few papers concerning its application for obtaining microparticles by the spray drying technique. The aim of this work was to compare the characteristics of microparticles made using EC in organic or aqueous form. The technological challenge was to reveal which type of EC is preferable for obtaining microparticles simultaneously characterized by appropriate pharmaceutical properties and a satisfactory degree of taste masking. Obtained microparticles were characterized in terms of morphology, encapsulation efficiency, yield, diameter size, Zeta potential, and, primarily, taste masking effect. In order to determine whether the designed microparticles effectively masked the taste of RUP, a human taste panel, the in vitro release, and a self-constructed electronic taste sensing system (electronic tongue) were used. 

## 2. Experimental Section

### 2.1. Materials

Ethocel^™^ and Surelease^®^ E-7-19040 was kindly donated from Colorcon Inc., Harleysville, PA, USA. Aquacoat^®^ ECD was a gift from FMC BioPolymer, Newark, NJ, USA. Mannitol (Parteck^®^ M) was received from Merck, Darmstadt, Germany. Ethanol was a product of POCH, Piekary Śląskie, Poland. RUP was obtained from Xi’An Kerui Biotechnology Co., Ltd., Xi’An, China.

### 2.2. Preparation of Microparticles Utilizing Organic Solution of EC

The feeding organic solutions were prepared by dissolving appropriate amounts of RUP and EC (Ethocel^®^) in 96% (*v*/*v*) ethanol. The microparticles were obtained using a Mini Spray Dryer B-290 (Büchi, Switzerland) equipped with an inert loop (B-295). To choose optimal spray drying parameters, a number of tests were conducted and the experimental parameters of the process were set as follows: inlet temperature 65 °C, aspirator flow 98%, feed flow 3.5 mL/min. All samples were prepared under identical drying conditions. 

To improve taste masking efficacy, mannitol addition as a flavor enhancer was examined. To optimize microparticle preparation, an experimental design approach was utilized. Twenty nine experiments (including 2 center points) (Table 1) were conducted to determine their optimal settings (SAS JMP program, version 8.0, SAS Institute, Cary, NC, USA). As crucial factors determining the quality of microparticles, encapsulation effectiveness, production yield, moisture content, and polydispersity index (PdI) were included. 

Design of experiments (DoE) methodology constitutes a powerful tool for systematic investigation of parameters influencing preparation of a given drug form or intermediate product. Central composite design, implemented in the presented study, enables delivery of predictions of high quality, while being robust to missing observations. Five factors, inlet temperature, spray rate, and mannitol concentration, as well as polymer concentration and its ratio to RUP, were tested. Six datasets, mass, production yield, moisture content, polydispersity index, percentage, and microparticle diameter, were included while building a model. All of the abovementioned responses were chosen due to the fact that they can be measured precisely and they provide useful information about the resulting formulations. However, as shown in Figure 1, despite the use of the maximize desirability function, allowing prediction of the optimal combination of factors’ settings (presented in Figure 1, in red above each factor), the optimal values (predicted by the model) were still quite average—not much better than those obtained with other settings in the tested ranges. 

Additionally, as can be seen in Figure 2, even a slight tightening of criteria for the output parameters caused the suitable ranges of input variables to become significantly smaller. Despite higher requirements for limits referring to values of responses (Figure 2b), there was still a small range of factor settings that enabled us to obtain the desired properties. This means that the best and most optimal values off selected responses can be achieved in such parameter settings, however, production of such a formulation will be characterized by insufficient robustness. On the basis of these results, it was concluded that formulations prepared within tested ranges of studied factors will be characterized by similar properties. Therefore, the essential factor of selecting a formulation should be its taste.

### 2.3. Preparation of Microparticles with Utilizing Aqueous Dispersions of EC

Based on the experiments with organic solutions of EC, it was established that the optimal concentration of EC is 6% and optimal RUP:EC ratio 1:2, therefore, those parameters were chosen. Microparticles with aqueous dispersions of EC (Surelease^®^ and Aquacoat^®^ ECD) were prepared using appropriate amounts of RUP suspended in the dispersions, and diluted with distilled water to obtain a 6% EC concentration (Table 2). The spray drying parameters were set as follows: inlet temperature 85 °C, aspirator flow 98%, feed flow 3.5 mL/min. To prevent settling of suspended particles, the suspensions were stirred throughout the whole process. The particles were collected after drying and kept in glass vessels. 

### 2.4. Microparticle Characterization

#### 2.4.1. Particle Size, Zeta Potential, and Morphology Evaluation

Measurements of the particle size, Zeta potential, and PdI were evaluated using a Zetasizer ZS 900 (Malvern Instruments, Malvern, UK). Morphology was analyzed by scanning electron microscope (SEM) (Inspect™S50, FEI Company, Hillsboro, OR, USA) at room temperature. Powder samples were attached to double-slide adhesive carbon tapes placed on SEM attachment and sputtered with gold. Measurements were taken in vacuum at different magnifications. 

#### 2.4.2. Determination of RUP Encapsulation Efficiency and Production Yield

To assess RUP loading, appropriate amounts of microparticles were weighted and dissolved in 10 mL 96% (*v/v*) ethanol and agitated for 30 min at 150 rpm in a water bath, then diluted with mobile phase. After filtration through 0.22 μm nylon syringe filters (Laboplus, Warszawa, Poland) RUP concentration was analyzed by HPLC (Agilent Technologies 1200, Agilent, Waldbronn, Germany) at 245 nm using a Waters Spherisorb 5μm ODS1 4.6 × 250 mm column (Waters Corporation, Milford, CT, USA). Phosphate buffer (pH = 3.0):methanol (65:35, *v/v*) was used as a mobile phase with 1.0 mL/min flow rate [26,27,28]. The retention time of RUP was 6.0 min. Standard calibration curve was linear over the range of 1–50 μg/mL, with the correlation coefficient R^2^ = 0.999. 

Mean drug encapsulation efficiency was assessed using the following equation:
*EE* = *Q*a/*Q*t × 100 (1)
where *EE* is percent of encapsulation efficiency, *Q*a is actual drug content and *Q*t is theoretical drug content.

The percent yield (Y) of RUP in microparticles was determined using the following equation:
*Y* = *m*m/*m*t × 100 (2)
where *m*m is a mass of microparticles and *m*t is theoretical mass of the drug and polymer. Yield was based on the percentage solid mass recovery from the spray dryer.

#### 2.4.3. Moisture Content

Moisture content in microparticles was assessed using a moisture analyzer balance (Radwag, Radom, Poland). 

### 2.5. Evaluation of Taste Masking Effectiveness

#### 2.5.1. In Vivo—Human Taste Panel

Evaluation of taste masking efficacy was performed in vivo in a human taste panel, with six healthy volunteers participating in a blind test (Research Ethics Committee at Medical University of Białystok approval number R-I-002/438). The study was carried out in the following stages: microparticles emplaced on a tongue (mass corresponding to 10 mg of RUP) for 30 s (maximal time when orodispersible drug dosage form should disintegrate in the oral cavity according to FDA directives [29]), drug spitting, mouth rinsing with water. The following scale was used to evaluate the taste: 0—not bitter, 1—slightly bitter, 2—moderately bitter, 3—very bitter. Prior to the test, the volunteers were selected by sensory sensitivity test, using basic flavors: sour—tartaric acid, sweet—sucrose, salty—sodium chloride, bitter—quinine hydrochloride [30]. 

#### 2.5.2. In Vitro RUP Release

The release of RUP from obtained microparticles was performed in a paddle apparatus (Erweka Dissolution Tester DT 600HH, Heusenstamm, Germany) using 40 mL of phosphate buffer pH = 6.8 simulating human saliva. Apparatus was constantly rotated at 75 rpm, bath temperature was 37 °C (+/−0.5). Sodium lauryl sulfate was added to dissolution medium to maintain sink conditions. The amount of released RUP was determined by HPLC, as described in point 2.4.2.

#### 2.5.3. Electronic Tongue

##### Membrane Materials

The following membrane components: polymer: poly(vinylchloride) (PVC) (Tarwinyl, Tarnów Poland); plasticizers: bis(2-ethylhexyl) sebacate (DOS), o-nitrophenyl octyl ether (o-NPOE) (Fluka, St. Gallen, Switzerland); lipophilic salts: potassium tetrakis[3,5-bis(trifluoromethyl)phenyl]-borate (KTFPB), tridodecylmethylammonium chloride (TDMAC) (Sigma-Aldrich, St. Luis, MO, USA), tributylhexadecylphosphonium bromide (TBHDPB) (Sigma-Aldrich, St. Luis, MO, USA); ionophores: heptyl 4-trifluoroacetyl-benzoate (ETH 6010, carbonate ionophore I) (Sigma-Aldrich, St. Luis, MO, USA), calix[6]arene-hexaacetic acid hexaethylester (amine ionophore I) (Fluka, St. Gallen, Switzerland) were used. All inorganic salts used were of analytical grade. All aqueous solutions were prepared with deionized water of conductivity 0.07 μs/cm (Elix Advantage System Mili-Q plus Milipore, Spittal an der Drau, Austria).

##### Membrane Preparation

The membrane phase of the electrode consists of two layers: the inner layer, containing plasticized PVC in which the Ag/AgCl electrode is placed, and the outer layer, contacting with the tested solution and containing the potential creating substance apart from the inner layer components. The internal layer consists of the following ingredients: 30% (*w/w*) PVC, 70% (*w/w*) of plasticizers, DOS, or o-NPOE. The components were mixed and the mixture was deaerated. The Teflon sensors were filled with the mixture, so that the Ag/AgCl electrode was immersed in it. Then the mixture was gelated at 373 K for 30 min. In order to prepare the outer layer, 27%–37% (*w/w*) PVC, 62%–68% (*w/w*) of plasticizers, and 3%–5% (*w/w*) of electroactive components were mixed (Table 3). The mixture was dissolved in tetrahydrofuran (THF) and applied dropwise on the inner layer, leaving THF to evaporate at 293 K, which was repeated several times. The electrodes prepared were stored in air between the measurements. 

##### Potentiometric Measurements

The sensor array consisted of 16 ion-selective electrodes with plasticized PVC membranes of solid contact architecture. All measurements were carried out in cells of the following type: Ag, AgCl; KCl 3M│CH_3_COOLi 1M│sample solution║membrane║PVC+plasticizer (solid contact); AgCl, Ag. The measurements of the electromotive force of the system ion-selective electrodes—reference electrode (Orion 90-02)—were carried out using Electrochemistry EMF Interface system (Lawson Labs. Inc., Malvern, PA, USA) and IBM PC computer. Before the first measurement, the electrodes were preconditioned at least 24 h in the adequate solutions. Regarding the poor solubility of (RUP) in water, the calibration curves of the constructed electrodes were examined by measuring the electromotive forces (EMF) in the concentration range of RUP 10^−5^–10^−3^ M. Before samples’ measurements, the signals of the electrodes were recorded in KNO_3_ solution (concentration range 10^−4^–10^−2^ M), in order to verify their proper operation. The signals of the sensor array were registered over 5 min in 5 repetitions. Between sample measurements, the sensors were washed and conditioned in 10^−3^ M NaCl solution for 5 min. Due to the very low solubility of the formulation in water, each solution of the heterogeneous sample was prepared separately and homogenized in the ultrasonic bath for 3 min immediately before the measurements.

##### Data Analysis

Chemical images of investigated formulations were obtained on the basis of steady-state responses of 16 potentiometric sensors (the 10 last readings of each sensor were averaged); therefore, each sample was characterized by 16 variables. The data matrix was processed using principal component analysis (PCA). All calculations and data analysis were performed in MatLab (The MathWorks Inc., Natick, MA, USA), SOLO^®^ software (Eigenvector Research Inc., Manson, WA, USA), and Origin (Microcal Software Inc., Northampton, MA, USA) software.

## 3. Results and Discussion

### 3.1. Characteristics of Microparticles Obtained from Organic EC Solution by Spray Drying

As many drugs are characterized by a bitter taste, overcoming this limitation is a great goal in designing pharmaceutical formulations. The spray drying technique can be applied to obtain polymer taste masked particles, which can be further incorporated into a variety of oral drug dosage forms. The process is relatively simple to carry out, however, obtaining the product with desired properties is a complex procedure, depending on many different factors. Most polymers used as taste masking agents are water-insoluble, thus, organic solvents are used, however, use of water as a solvent is preferable because of lower costs and nontoxicity. In the first stage, a drug is dissolved or suspended into polymer solution. Next, liquid is drawn by the peristaltic pump and sprayed into the drying chamber in microdrop form. This is an important moment in whole process, as it determines the size of obtained droplets, hence it affects the speed and efficiency of drying in the further phases and in consequence the size of the microparticles. In the next stage, drops are dried in a hot drying gas, nitrogen. Obtaining a large area of sprayed liquid facilitates the heat exchange from drying gas to liquid particles, so the solvent evaporates very quickly and the drug is incorporated into the polymer shell. As the degree of solvent evaporation increases, the diameter of the dried drop decreases and the concentration of solid substances on its surface increases. The final effect of the process is microparticles accumulated in the bottom of the collector. It should be emphasized that, although drying takes place in a stream of hot gas, the temperature values reached are definitely below temperatures used in other methods of obtaining microparticles, which allows for drying of temperature-sensitive materials, maintaining their activity [5]. To formulate microparticles with RUP using an organic EC solution, parameters of the process were set as follows: inlet temperature 65 °C, aspirator flow 98%, and feed flow 3.5 mL/min. The characteristics of the microparticles are given in Table 4. 

The particle size was in the range from 1.2 μm to 4.9 μm, and no dependence between particle size and drug:polymer ratio was observed. Formulation F_18_, with the largest encapsulation efficiency (99.6%) had a mean diameter of 1.8 μm, and formulation F_2_, with the lowest encapsulation efficiency, about 3.8 μm. The lowest production yield was observed for F_1_ (10.7%), while the highest was for F_9_ (79.5%). Similar values of production yield (up to 70%) and drug-loading efficiency (up to 96%) were noted by Garekani et al. in preparation of microparticles from organic EC solutions containing theophylline [31]. Interestingly, they observed significantly larger particles (even up to 25 µm) compared to the particles obtained in this study. 

Moisture content in all designed formulations was in the range of 0.1–11.3%. Moisture content is an essential indicator of spray drying effectiveness, and thus product quality. It affects the flow properties, compressibility, and stability or tendency to agglomerate of the obtained product. Values of polydispersity index (PdI) from 0.08 to 0.7 are defined as mid-range, whereas values 0.7–1.0 indicate a very broad distribution of particle size, and large particles or aggregates that could be slowly sedimenting. In the case of pharmaceutical microparticles, a PdI below 0.3 is desired, however, practically its value should be less than 0.5 [32]. From the obtained results, it can be seen that the PdI values differed from each other, ranging from 0.2 (F_5_) to 1.0 (F_8,_ F_13,_ F_14,_ F_15,_ F_17_). Zeta potential is a key indicator of the colloidal dispersion’s stability and is a measure of particle charge magnitude—the higher the value (positive or negative), the more stable the dispersion is. For molecules and particles that are small enough, a high Zeta potential confers stability, i.e., the solution or dispersion will resist aggregation. When the potential is small, attractive forces may exceed this repulsion and the dispersion may break and flocculate. Correlation between Zeta potential value and stability shows that a value of minimum > +/−30 mV indicates good stability. In many cases, a change in Zeta potential is also an indicator of the extent of drug loading in the particles. Higher Zeta potential values are necessary to ensure stability and to avoid aggregation of particles [33]. The Zeta potentials of all obtained formulations reached very low values, which might suggest instability or a tendency to aggregate.

### 3.2. In Vivo Taste Evaluation

In order to determine the taste masking effect of microparticles designed from organic EC solution, a human taste panel was applied as a preliminary screening test. Taste was initially examined by six selected, healthy volunteers. Organoleptic analysis of microparticles revealed that formulations obtained from organic solutions of EC (F_1_–F_9_) were rated as very or moderately bitter (only one formulation—F_4_—was assessed as moderately, slightly bitter, or not bitter). Surprisingly, mannitol addition only barely improved the taste masking effect (formulation F_13_) (Table 5). 

### 3.3. Characteristics of Microparticles Obtained from Aqueous EC Dispersions

Microparticles obtained from organic solutions of EC, even with MN addition, were characterized by an unsatisfying taste masking effect, which is a crucial parameter in designing pharmaceutical formulations. Therefore, in the next stage, microparticles prepared from aqueous EC dispersions were evaluated (Table 6). The size of particles was very similar (from mean value 3.2 μm for F_19_ to 3.6 μm for F_20_), and both formulations were characterized by high encapsulation efficiency (86.1% and 95.1% in F_19_ and F_20,_ respectively). Interestingly, moisture content in the obtained formulations was significantly different (2.5% in F_19_ and 8.6% in F_20_). In both formulations, PdI was smaller than 0.5. Similarly, as was observed in formulations prepared from organic EC solutions, the Zeta potential reached very low values. Regarding the taste masking effectiveness, all probands indicated formulations F_19_ and F_20_ as 0 (not bitter). Moreover, the volunteers did not report feeling any other unpleasant organoleptic sensations. 

Formulations F_4_, F_13_, F_19_, and F_20_—with the best taste masking efficiency evaluated in vivo by volunteers—were subjected to further, more detailed analysis using two additional methods: in vitro release test and electronic tongue.

### 3.4. In Vitro RUP Release and Morphology Evaluation

Slower drug release is significantly correlated with taste masking effectiveness [30]. RUP release from formulations F_4_ and F_13_ was very rapid (Figure 3), and thus their taste was masked only to a minor extent. SEM photomicrographs revealed that microparticles prepared with organic solutions (F_4,_ F_13_) showed a collapsed, wrinkled, irregular structure with small particles adsorbed on external surface, and they resembled “shriveled raisins” (Figure 4). When the particles were exposed to the dissolution medium, the drug present on the outer surface released suddenly. The particles obtained from aqueous EC dispersions proved to be mainly spherical, homogenous, smooth surfaced (Figure 4), and they released RUP significantly slower. After 3 min of the dissolution test, about 30% and 6% of RUP was released from F_19_ and F_20_, respectively (Figure 3). The shape of the particles obtained from the aqueous dispersions was probably influenced by less surface tension, affected by additional excipients present in the dispersions. In both of them, stabilizers, plasticizers, and surfactants were used (oleic acid and fractionated coconut oil in Surelease^®^, cetyl alcohol and sodium lauryl sulfate in Aquacoat^®^). Garekani et al. attempted to compare aqueous EC dispersion (Surelease^®^) and organic EC solution (Ethocel^®^), with an EC final concentration of 7.5% in the preparation of sustained release theophylline microparticles. Dissolution studies were performed using a USP apparatus II with 1000 mL of distilled water. The results showed that aqueous samples released 50% of drug in less than 1 hour, and about 15% within 5 min, which is comparable to our results. Contrarily, organic samples showed significantly slower theophylline release [31]. However, it should be emphasized that the study was carried out in a different medium and in larger volume (1000 mL of water versus 40 mL of phosphate buffer pH = 6.8). Based on SEM pictures, they revealed that the microparticle structure was affected by the type of vehicle used during the spray drying. On the surfaces of particles obtained from ethanolic EC solution, fewer drug crystals were observed, suggesting theophylline encapsulation within particles [31]. Emami et al. designed EC microspheres with theophylline to prepare sustained release oral suspension. Only in case of EC solution in methylene chloride were spherical microcapsules with sustained release effect obtained. To determine theophylline release, the USP paddle method and 900 mL of phosphate buffer pH = 4.5 as dissolution medium was applied. After 25 min, about 10% of the drug was released. Minimal burst effect was observed at the early stage, related to the broken particles or drug crystals not engulfed with the polymer. Microparticles obtained using other solvents (ethanol/water, acetone, ammonium chloride) were not spherical, broken, had rough surfaces, and thus released the drug very quickly within the first 20 min [34]. Rattes et al. used aqueous EC dispersion (Surelease^®^) to sustain the release of sodium diclofenac from microparticles. By utilizing the paddle method with 750 mL of 0.1 M HCl followed by 1000 mL of phosphate buffer pH = 6.8, they revealed delayed drug dissolution rates [35]. Arici et al. tested aqueous EC dispersion (Aquacoat^®^ ECD) in preparation of naproxen microparticles. The release study was performed in 100 mL of phosphate buffer pH = 7.4. For formulation with a drug:EC ratio of 1:4, about 10% of a drug was released after 5 min, and about 45% after 30 min. SEM photomicrographs revealed spherical, well-separated particles with a smooth surface and absence of agglomerates [36].

### 3.5. Electronic Tongue

#### 3.5.1. Performance of the Potentiometric Sensors

The sensor array used in the electronic tongue system included eight types of ion-selective electrodes with solid contacts, with two sensors of each type. Before the measurements of microparticle formulations, the calibration curves of the electrodes towards RUP and other excipients such as EC and MN were determined. On the basis of the results shown in Figure 5, it was concluded that the cation selective electrodes achieved the highest sensitivity (about 70 mV decade^−1^), and the electrode with amine ionophore, near Nernstian response, 61 mV decade^−1^. Furthermore, lower sensitivities were recorded for electrodes based on ammonium and phosphonium lipophilic salts. The slope of the linear range of calibration curves within limits of 30–35 mV decade^−1^ for AS-D and AS-N electrodes and 10–36 mV decade^−1^ for PS-D and PS-N electrodes, respectively. The noticeable sensitivity of the presented electrodes can be explained by the ionic nature of RUP (two carboxyl groups) and the interaction with salt molecules or ionophores in the membrane.

There were significant differences in the response of the electrodes in the excipient solutions: EC, MN. The sensitivity toward these compounds was lower compared to pure RUP due to their non-ionic character. The electrode AM-D exhibited a linear range, with a slope of about −10 mV decade^−1^. The other sensors practically did not show sensitivity towards this polymer. Different cationic or anionic sensitivity was obtained for electrodes relative to MN. The electrode CS-D displayed relatively slightly higher sensitivity towards mannitol. Lower sensitivity was obtained for sensors AS-D and PS-N, while the others did not show practical sensitivity. However, there was a clear difference in the response of the developed electrodes towards the studied substances, which made it possible to apply them as an electronic tongue sensor array for the analysis of pharmaceutical formulations based on RUP. 

#### 3.5.2. Electronic Tongue Results

The electronic tongue was applied to study differences of the chemical images of the following samples: formulations F_4_, F_13,_ F_19_, F_20_, as well as respective placebos, pure RUP, and pure EC and MN. Every sample was measured in 5 replicates, which resulted in a 60 × 16 data matrix. These data were processed using principal component analysis (PCA), prior to which autoscaling was applied. 

The final chemical images were presented on 2D-PCA score plots (Figure 6). They revealed that all studied samples exhibited repeatable responses, due to formation of respective, compact clusters for each sample type. Pure RUP was easily discernable from all other samples and was characterized by the highest value of PC2 (Figure 6a) or PC1 (Figure 6b). All placebos were linearly separable from RUP, as well as from formulations and pure excipients, moreover, they formed separate clusters (all placebos were similar in terms of electronic tongue response). The most distant from pure RUP were all placebos and pure excipients (Figure 6a), while all formulations F_4_, F_13_, F_19_, F_20_, were placed between RUP and the placebo cluster, showing intermediate properties between RUP and their respective placebos. A masking effect was visible when distinct clusters are observed on PCA score plot, and this was the case (all formulation clusters were linearly separable from points representing the pure drug, therefore the electronic tongue can detect the difference between pure RUP and RUP microparticles). 

To study differences between the studied formulations in detail, PCA was repeated for a data matrix in which the pure excipients were excluded (Figure 6b). Appropriate classification accuracy of the electronic tongue used in this study was observed again; moreover, all formulations, as expected, were again placed between RUP and the cluster formed by all placebos. It must be noted that two formulations, F_4_ and F_13_, showed an almost identical chemical image on both PCA score plots; their clusters were not separable in PC1–PC2 space, showing 83%–87% of the variance of the data set. This observation is in good agreement with both release studies (Figure 3) as well as in vivo sensory evaluation. Their placebos are also hardly discernable, due to the fact that they are both based on the same polymer (EC). The most similar to its placebo was formulation F_20_, suggesting the best taste masking efficiency, which correlates with the release study (Figure 3). However, in the electronic tongue results, the F_19_ chemical image was the most similar to RUP from all studied formulations. This can be related to the fact that the respective placebo also gave the most similar response to the pure drug from all placebos (F_19_ placebo cluster was closer to RUP than F4 placebo, F_13_ placebo, and F_20_ placebo). In these terms, the relative distance to RUP for formulations F_4_, F_13_, and F_19_ is comparable. All these findings confirm that proposed electronic tongue system can be applied to study taste masking effects in microparticles based on RUP. 

## 4. Conclusions

One of the novel techniques for masking unpleasant tastes of drugs is designing polymer microparticles. By using the spray drying method to obtain microparticles with masked extremely bitter RUP, it was shown that EC in the aqueous dispersion form is preferable. Surprisingly, mannitol addition only barely improved the taste masking effect of formulations obtained from organic EC solution. The microparticles obtained from aqueous dispersions of EC proved to be of better quality in terms of morphology, and, most essentially, taste masking efficacy. Utilizing three independent taste assessment methods: in vivo with healthy volunteers’ participation, in vitro release, and a self-constructed electronic tongue it was demonstrated that microparticles formulated from aqueous dispersions of EC provided a very effective taste masking barrier. Designed microparticles containing RUP seem to be promising carriers and might be further used to formulate dosage forms with a taste masked effect. However, possibilities of their exploitation in liquid and solid pharmaceutical formulations have to be precisely examined.

## Figures and Tables

**Figure 1 polymers-11-00522-f001:**
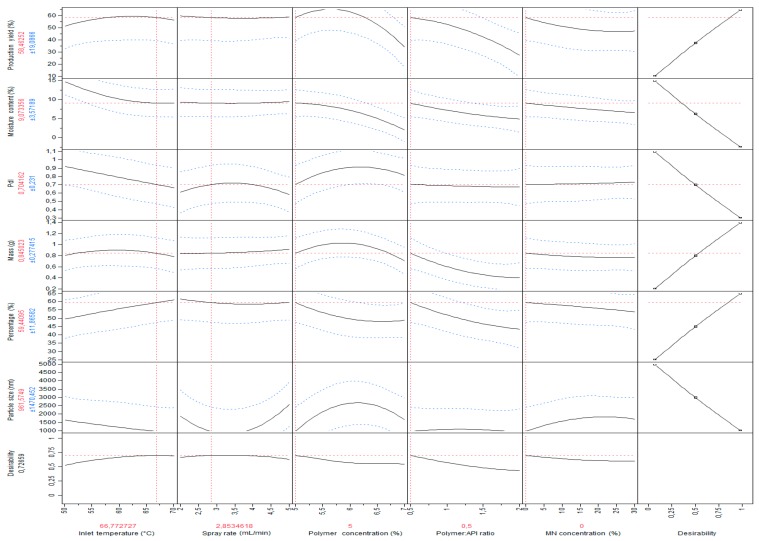
Prediction profiler with maximize desirability function used.

**Figure 2 polymers-11-00522-f002:**
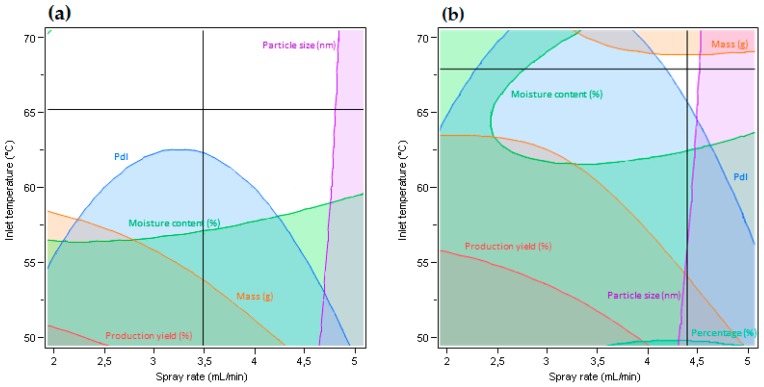
Contour plot function for determining ranges of settings for two selected factors (uncolored space) in order to obtain desired values of responses. (**a**) Production yield ≥ 30 %, moisture content ≤ 8%, polydispersity index ≤ 0.8, mass ≥ 0.6 g, percentage ≥ 30 %, microparticle diameter ≤ 3500 nm; (**b**) production yield ≥ 40 %, moisture content ≤ 7%, polydispersity index ≤ 0.7, mass ≥ 0.7 g, percentage ≥ 40 %, microparticle diameter ≤ 3000 nm).

**Figure 3 polymers-11-00522-f003:**
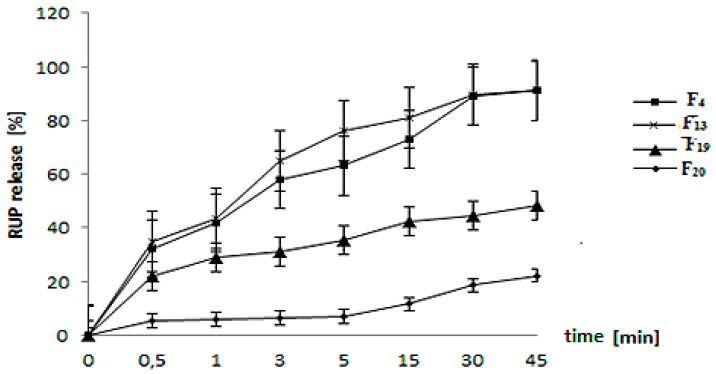
Rupatadine fumarate (RUP) release from formulations F_4_, F_13_, F_19_, and F_20_ in phosphate buffer pH = 6.8.

**Figure 4 polymers-11-00522-f004:**
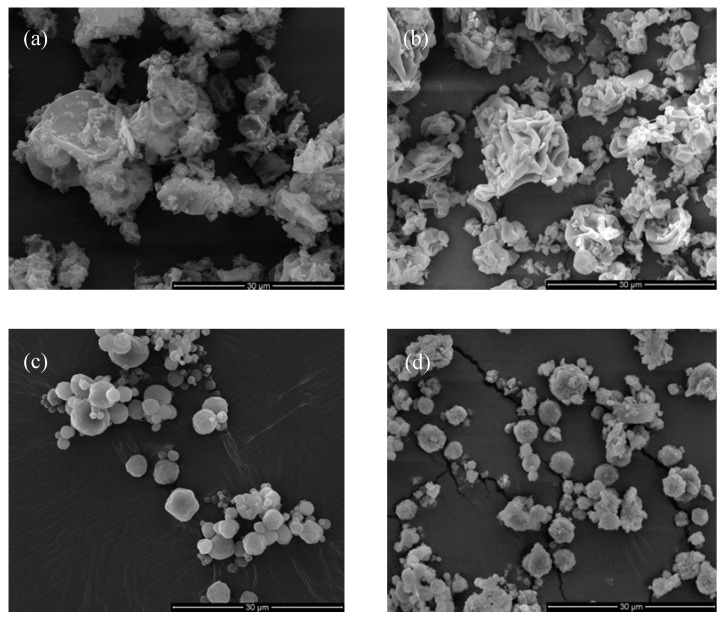
Scanning electron microscope (SEM) pictures of formulations: (**a**) F_4_, (**b**) F_13,_ (**c**) F_19_, (**d**) F_20_ under 5000× magnification.

**Figure 5 polymers-11-00522-f005:**
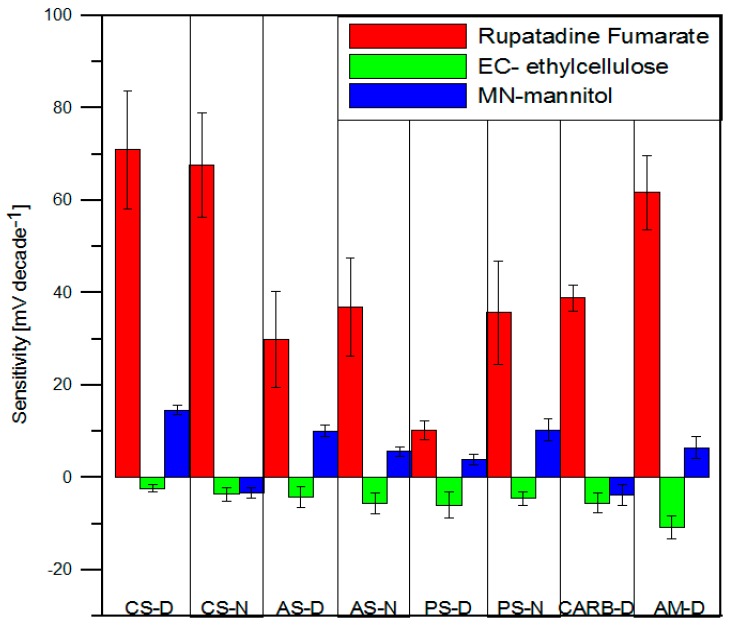
Sensitivity of sensors towards RUP, EC, MN.

**Figure 6 polymers-11-00522-f006:**
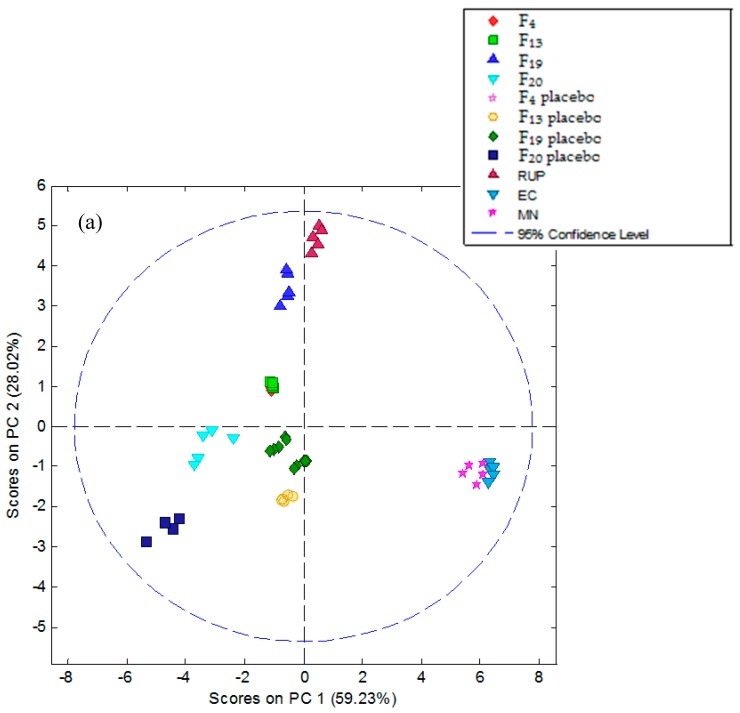
Principal component analysis (PCA) plots presenting taste clusters of: pure RUP, EC, MN, microparticle placebo and microparticle formulations F_4_, F_13_, F_19_, and F_20_ (**a**) and pure RUP, microparticle placebo and microparticle formulations F_4_, F_13_, F_19_, and F_20_ with the pure excipients exclusion (**b**).

**Table 1 polymers-11-00522-t001:** The plan of experiments concerning preparation of formulations from ethylcellulose (EC) organic solutions with mannitol addition.

Formulation	Pattern	Inlet Temperature (°C)	Spray Rate (mL/min)	Polymer Concentration (%)	Polymer:API Ratio	Mannitol (MN) Concentration (%)
1	+−+−−	70	2	7	0.5	0
2	+++−+	70	5	7	0.5	30
3	A0000	70	3.5	6	1.25	15
4	00000	60	3.5	6	1.25	15
5	−−++−	50	2	7	2	0
6	00a00	60	3.5	5	1.25	15
7	+−+++	70	2	7	2	30
8	−−+−+	50	2	7	0.5	30
9	−+−+−	50	5	5	2	0
10	+−−−+	70	2	5	0.5	30
11	−++++	50	5	7	2	30
12	++++−	70	5	7	2	0
13	−−−−−	50	2	5	0.5	0
14	000A0	60	3.5	6	2	15
15	+−−+−	70	2	5	2	0
16	−−−++	50	2	5	2	30
17	++−−−	70	5	5	0.5	0
18	0a000	60	2	6	1.25	15
19	0A000	60	5	6	1.25	15
20	00A00	60	3.5	7	1.25	15
21	−++−−	50	5	7	0.5	0
22	0000a	60	3.5	6	1.25	0
23	a0000	50	3.5	6	1.25	15
24	0000A	60	3.5	6	1.25	30
25	00000	60	3.5	6	1.25	15
26	000a0	60	3.5	6	0.5	15
27	−+−−+	50	5	5	0.5	30
28	++−++	70	5	5	2	30
29	00000	60	3.5	6	1.25	15

**Table 2 polymers-11-00522-t002:** Composition of designed formulations obtained with organic solutions of EC (F_1_–F_9_), with mannitol addition (F_10_–F_18_) and with aqueous EC dispersions (F_19_–F_20_).

Formulation Number	EC Concentration	Drug:Polymer Ratio	MN Concentration
**Formulations with organic solution of EC**			
F_1_	5%	1:2	-
F_2_	5%	1:1	-
F_3_	5%	2:1	-
F_4_	6%	1:2	-
F_5_	6%	1:1	-
F_6_	6%	2:1	-
F_7_	7%	1:2	-
F_8_	7%	1:1	-
F_9_	7%	2:1	-
**Formulations with organic solution of EC with MN addition (based on DoE)**			
F_10_	5%	1:2	30%
F_11_	5%	1.25:1	15%
F_12_	5%	2:1	30%
F_13_	6%	1:2	15%
F_14_	6%	1.25:1	15%
F_15_	6%	2:1	15%
F_16_	7%	1:2	30%
F_17_	7%	1.25:1	15%
F_18_	7%	2:1	30%
**Formulations with aqueous dispersions of EC**			
F_19_ *	6%	1:2	-
F_20_ **	6%	1:2	-

* Surelease^®^ E-7-19040, ** Aquacoat^®^ ECD.

**Table 3 polymers-11-00522-t003:** Components used for sensor membranes preparation.

Electrode Number	Electrode Type	Ionophore (% *w*/*w*)	Lipophilic Salt (% *w*/*w*)	Plasticizer (% *w*/*w*)	Polymer (% *w*/*w*)
1–2	CS-D	-	KTFPB (1%)	DOS (66%)	PVC (33%)
3–4	CS-N	-	KTFPB (1%)	NPOE (66%)	PVC (33%)
5–6	AS-D	-	TDMAC (3.5%)	DOS(64%)	PVC (32.5%)
7–8	AS-N	-	TDMAC (3.5%)	NPOE (64%)	PVC (32.5%)
9–10	PS-D	-	TBHDPB (3.5%)	DOS (64%)	PVC (32.5%)
11–12	PS-N	-	TBHDPB(3.5%)	NPOE(64%)	PVC (32.5%)
13–14	CARB-D	ETH 6010 (0.7%)	TDMAC (0.3%)	DOS (62%)	PVC (37%)
15–16	AM-D	amine ionophore I (5%)	-	DOS (68%)	PVC (27%)

**Table 4 polymers-11-00522-t004:** Characteristic of obtained microparticles (F_1_–F_18_).

Formulation Number	Encapsulation Efficiency %	Production Yield %	Moisture Content (%)	Polydispersity Index	Zeta Potential (mV)	Microparticle Diameter (μm)
F_1_	62.2	10.7	5.8 ± 2.4	0.4 ± 0.2	0.1 ± 1.5	2.9 ± 2.0
F_2_	41.6	60.5	4.0 ± 2.9	0.4 ± 0.4	0.1 ± 0.4	3.8 ± 0.9
F_3_	47.4	58.9	2.8 ± 1.2	0.6 ± 0.5	−0.8 ± 0.5	4.0 ± 0.7
F_4_	75.3	57.2	2.6 ± 0.8	0.3 ± 0.3	0.6 ± 1.2	2.3 ± 0.9
F_5_	57.2	40.5	10.7 ± 2.1	0.2 ± 0.4	−0.3 ± 0.6	2.7 ± 1.5
F_6_	47.4	33.4	8.9 ± 1.6	0.3 ± 0.1	−0.5 ± 0.6	2.4 ± 0.4
F_7_	72.6	42.2	5.4 ± 0.7	0.4 ± 0.3	0.5 ± 1.8	2.7 ± 0.6
F_8_	51.5	62.1	8.3 ± 2.2	1.0 ± 0.0	0.3 ± 1.1	1.2 ± 0.2
F_9_	64.9	79.5	6.8 ± 1.1	0.9 ± 0.3	0.8 ± 0.5	1.8 ± 0.9
F_10_	68.9	39.2	11.2 ± 2.4	0.7 ± 0.1	0.1 ± 0.2	4.6 ± 0.5
F_11_	98.9	40.1	4.9 ± 0.2	0.7 ± 0.3	−0.1 ± 0.2	1.7 ± 0.8
F_12_	99.4	50.4	6.9 ± 1.2	0.9 ± 0.2	0.1 ± 0.2	4.5 ± 0.1
F_13_	73.9	37.0	7.3 ± 4.3	1.0 ± 0.0	−0.2 ± 0.1	4.9 ± 2.2
F_14_	97.5	27.4	9.5 ± 1.2	1.0 ± 0.0	−0.2 ± 0.4	2.6 ± 0.1
F_15_	99.0	49.3	7.7 ± 0.6	1.0 ± 0.0	−0.2 ± 0.1	3.1 ± 0.2
F_16_	92.7	32.7	0.1±0.3	0.5 ± 0.3	−0.4 ± 0.2	1.9 ± 0.3
F_17_	91.1	23.3	11.2 ± 2.6	1.0 ± 0.0	0.2 ± 0.4	1.4 ± 0.1
F_18_	99.6	12.8	11.3 ± 1.3	0.9 ± 0.1	−0.2 ± 0.1	1.8 ± 0.1

**Table 5 polymers-11-00522-t005:** Sensory evaluation of microparticles obtained with organic EC solution (formulations F_1_–F_9)_ and with MN addition (formulations F_10_-F_18_), scored as follows: 0—not bitter, 1—slightly bitter, 2—moderately bitter, 3—very bitter.

Formulation and Score																			
		F_1_	F_2_	F_3_	F_4_	F_5_	F_6_	F_7_	F_8_	F_9_	F_10_	F_11_	F_12_	F_13_	F_14_	F_15_	F_16_	F_17_	F_18_
**Volunteer**	A	2	2	3	2	3	3	2	2	3	2	2	3	2	2	2	2	2	2
	B	1	2	2	1	2	2	0	1	1	1	2	3	1	1	2	1	2	2
	C	1	2	2	1	3	3	1	1	1	2	1	2	1	2	1	2	2	2
	D	3	3	3	2	3	3	2	3	3	1	1	2	1	1	2	2	1	3
	E	1	2	3	0	1	2	1	2	3	2	2	2	1	0	1	1	2	2
	F	2	3	3	0	2	3	1	2	3	2	3	2	0	1	2	2	2	2

**Table 6 polymers-11-00522-t006:** Characteristics of microparticles obtained from aqueous EC dispersions (F_19_ and F_20_).

Formulation Number	Encapsulation Efficiency %	Production Yield %	Moisture Content (%)	Polydispersity Index	Zeta Potential (mV)	Microparticle Diameter (μm)
F_19_	86.1	76.2	2.5 ± 0.4	0.3 ± 0.1	0.3 ± 0.4	3.2 ± 1.1
F_20_	95.1	81.4	8.6 ± 0.5	0.4 ± 0.1	0.6 ± 0.4	3.6 ± 0.5

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
