# Peer review of "Ethylcellulose in Organic Solution or Aqueous Dispersion Form in Designing Taste-Masked Microparticles by the Spray Drying Technique with a Model Bitter Drug: Rupatadine Fumarate"

_polymers, 2019, doi:10.3390/polym11030522_

Round 1

Reviewer 1 Report

The whole article has to be improved.

The introduction has to be rewritten since there are descriptions like the spray-drying method that is not for the introduction but rather for the methods section.

Figure 1 is a table not a figure! The table must be improved and simplified. The last 4 columns must be deleted and mention on the text the evaluated parameters.

Figure 2 and Figure 3 must be improved to facilitate the reading.

The results are presented; the discussion is missing referring to previously published works.
For example the drug release curves are shown but not compared to the literature.

Very few bibliographical references are presented.

Author Response

Response to the Reviewer 1

According to the suggestion of the Reviewer, the introduction has been improved. Some information were deleted, some information about the spray drying method were moved into other section, and some information concerning ethylcellulose utilization in pharmaceutical technology was added as follow [page 2, lines 58-62]: “EC is widely used in the pharmaceutical technology, mostly in oral formulations. It is primarily employed as a coating agent modifying drug release, improving drug stability or providing moisture protection. EC is also utilized in modified-release tablets as an excipient creating matrix structure.” New references [11-16] were added to this section.

According to the suggestion of the Reviewer, Figure 1 was rearranged into Table 1. The last 4 columns were deleted and evaluated parameters were mentioned in the text as follow [page 3, lines 116-118]: “As crucial factors determining the quality of microparticles, encapsulation effectiveness, production yield, moisture content and polydispersity index (PdI) were included”.

According to the suggestion of the Reviewer, figures were corrected and their quality was improved. The information presented in Figure 2 were simplified and the caption has been changed as follow: “Figure 2. Contour plot function for determining ranges of settings for two selected factors (uncolored space) in order to obtain desired values of responses. (a: production yield ≥ 30 %, moisture content ≤ 8%, polydispersity index ≤ 0.8, mass ≥ 0.6 g, percentage ≥ 30 %, microparticles diameter ≤ 3500 nm; b: production yield ≥ 40 %, moisture content ≤ 7%, polydispersity index ≤ 0.7, mass ≥ 0.7 g, percentage ≥ 40 %, microparticles diameter ≤ 3000 nm).”

According to the suggestion of the Reviewer, the discussion was improved and extended:

1. Some information about the spray drying process from the introduction were shortened and moved [page 8, lines 292-307]; some information about significance of moisture content was added [page 9, lines 322-324]

2. Discussion referring to previously published works was added as follow:

- [page 9, lines 318-321] „Similar values of production yield (up to 70%) and drug-loading efficiency (up to 96%) were noted by Garekani et al. in preparation of

microparticles from organic EC solutions containing theophylline [31]. Interestingly, they observed significantly larger particles (even up to 25 μm) compared to particles obtained in this study.”

- [page 9, lines 388-413] “Garekani et al. attempted to compare using aqueous EC dispersion (Surelease®) and organic EC solution (Ethocel®) with EC final concentration 7.5% in preparation of sustained release theophylline microparticles. Dissolution studies were performed using USP apparatus II with 1000 mL of distilled water. The results showed that aqueous samples released 50% of drug in less than 1 hour and about 15% within 5 min, what is comparable to our results. Contrary, organic samples showed significantly slower theophylline release [31]. However, it should be emphasized that the study was carried out in different medium and in larger volume (1000 mL of water versus 40 mL of phosphate buffer pH=6.8). Based on SEM pictures, they have revealed that microparticles structure was affected by the type of vehicle used during the spray drying. On the surfaces of particles obtained from ethanolic EC solution, fewer drug crystals were observed suggesting theophylline encapsulation within particles [31]. Emami et al. designed EC microspheres with theophylline to prepare sustained release oral suspension. Only in case of EC solution in methylene chloride, spherical microcapsules with sustained release effect were obtained. To determine theophylline release, USP paddle method and 900 mL of phosphate buffer pH=4.5 as dissolution medium was applied. After 25 min, about 10% of drug was released. Minimal burst effect was observed at the early stage related to the broken particles or drug crystals not engulfed with the polymer. Microparticles obtained with using other solvents (ethanol/water, acetone, ammonium chloride) were not spherical, broken, having rough surface, thus released the drug very quickly within the first 20 min [34]. Rattes et al. used aqueous EC dispersion (Surelease®) to sustain the release of sodium diclofenac from microparticles. By utilizing paddle method with 750 mL of 0.1 M HCl followed by 1000 mL of phosphate buffer pH=6.8 they revealed delayed drug dissolution rates [35]. Arici et. al tested aqueous EC dispersion (Aquacoat® ECD) in preparation of naproxen microparticles. The release study was performed in 100 mL of phosphate buffer pH=7.4. For formulation with drug:EC ratio 1:4, about 10% of a drug was released after 5 min and about 45% after 30 min. SEM photomicrographs revealed spherical, well-separated particles with smooth surface and absence of agglomerates [36].”

According to the Reviewer, new bibliographical references were added (11-16; 31, 34, 35, 36).

We thank the Reviewer for the revision and these insightful suggestions. We find Reviewer suggestion to be very helpful and constructive and the corresponding revisions have strengthened the paper.

Reviewer 2 Report

The authors have designed the taste-masking microparticles by spray-drying with use of DOE method.  The concept of the study is interesting and well designed.  I recommend the authors to improve the appearance of figures before the publication.

Fig. 1-3,  These figures looks just screenshot of the software.  It is very difficult for readers to see and understand the contents.  Also, there might be too much information which are not important for the discussion.  It is confusing.  Please reconsider the contents of table and figure, and rearrange the figures and tables.

Fig. 5,  In image (a), there is a careless mistake (“(b)” is appeared in figure 5(a))

Fig. 6,  The resolution of the figure should be improved.  If it is difficult to improve the resolution, it is better to use larger font.

Author Response

Response to the Reviewer 2

- We agree with the Reviewer that Figures 1-3 were difficult for readers to see and understand the contents. Therefore, Figure 1 was rearranged into Table 1, quality of Figure 2 was improved and Figure 3 was simplified (new Figures 1-2).

- According to the suggestion of the Reviewer, Figure 5 was corrected.

- According to the suggestion of the Reviewer, the resolution of Figure 6 was improved to make it more readable.

We thank the Reviewer for the revision. Technical remarks of the Reviewer were considered and corrected. We apologize for all mistakes and misspellings.

Round 2

Reviewer 1 Report

My appreciation is accepted , since the authors have made all the addresses suggestions of my first revision.